# Learning affective meanings that derives the social behavior using Bidirectional Encoder Representations from Transformers

## Abstract

Predicting the outcome of a process requires modeling the system dynamic and observing the states. In the context of social behaviors, sentiments characterize the states of the system. Affect Control Theory (ACT) uses sentiments to manifest potential interaction. ACT is a generative theory of culture and behavior based on a three-dimensional sentiment lexicon. Traditionally, the sentiments are quantified using survey data which is fed into a regression model to explain social behavior. The lexicons used in the survey are limited due to prohibitive cost. This paper uses a fine-tuned Bidirectional Encoder Representations from Transformers (BERT) model to develop a replacement for these surveys. This model achieves state-of-the-art accuracy in estimating affective meanings, expanding the affective lexicon, and allowing more behaviors to be explained.

## 1 Introduction

Consider talking to your mentor for some advice about how to *behave* with your colleague. Your mentor probably starts asking questions about the *culture* in the workspace and may continue asking about the *identity* of the person. These questions could be about *institutional constraints* such as being the manager, or they could be personality *sentiment* such as being nice or active. Knowing this information, your mentor may have some initial recommendations, but you *adapt* your behavior after observing the reactions from the colleague. This is a descriptive scenario for a daily interaction. Affect Control Theory (ACT) quantifies the variables in this scenario and produces equations for simulating human behavior in social interactions.

ACT was introduced in the 1970s (Heise, 1977) and has been validated in more than 100 social science projects (Robinson & Smith-Lovin, 2018). ACT has been used in interdisciplinary applications such as Human-Computer Interactions (Robillard & Hoey, 2018), finding how language cultures affect social response (Kriegel et al., 2017), and modeling identities and behaviors within groups (Rogers & Smith-Lovin, 2019). More recently, Mostafavi (2021) introduced ACT to estimate and track emotional states during online messaging. For example, chatbots can use the ACT framework to understand the emotional state of the customer in real time and adapt their behavior accordingly. While the potential uses of ACT are understanding emotional changes during online messaging Mostafavi (2021), real life applications are limited due to the vocabulary size of affective dictionaries.

ACT uses a three-dimensional affective meaning space as a quantified form of sentiments (Heise, 1977). ACT uses Evaluation [good vs. bad], Potency [powerful vs. powerless], and Activity [active vs. passive] (EPA) space introduced by Osgood et al. (1957) as a semantic differential form of affective meaning. These affective lexicons represent the word of interest within cultural and social boundaries (Robinson & Smith-Lovin, 1992). Fontaine et al. (2007) found that EPA scores are the first three principal components after reducing dimensionality on 144 features representing the main components of emotions.

Historically, surveys are used to quantify the affective meanings within a cultural group. To compensate for unreliability in the survey, most EPA surveys are designed so that each word is scored by at least 25 different participants. Thus, finding the affective meaning for 5000 words requires over 125,000 ratings and 400 hours of respondent time (Heise, 2010). Because ACT also requires

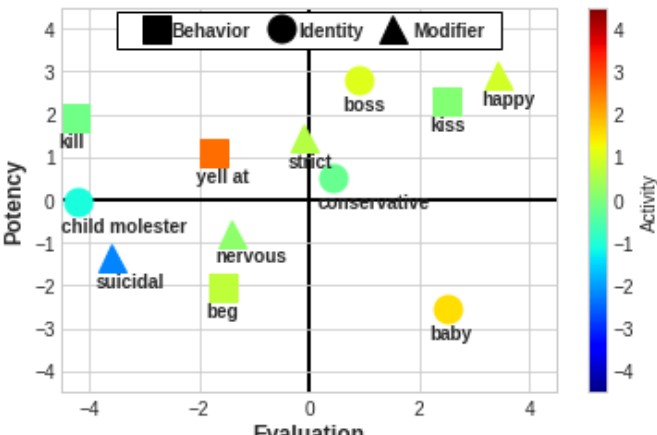

Figure 1: Sentiments of sample words in EPA space. Circles, squares, and triangles show identities, behaviors, and modifiers. The color represents the activity dimension. Comparing words such as "happy" and "suicidal" we can observe one has positive values in all three dimensions and the other has negative values.

EPA estimates for social interactions, EPA surveys must also include participants to score a large number of additional scenarios. Due to the high cost and time required, most EPA surveys have been focused on a small number of words which has limited the applicability of ACT. As an alternative to conducting new surveys, researchers have tried supervised (Mostafavi & Porter, 2021; Li et al., 2017) and semi-supervised (Alhothali & Hoey, 2017) methods on shallow word-embeddings to find affective lexicons. However, their performance on the *activity* and *potency* dimensions is poor. In this paper, we discuss the main limitations for the applicability of shallow word-embeddings. In this work, we use deep sentence-embedding to consider contexual aspects of concepts in social events. For that purpose, we show how to generate a contextual data-set describing social events to train and test a deep neural network. We use Bidirectional Encoder Representations from Transformers (BERT) embedding tuned for finding affective lexicons. The performance of our approach achieves state-of-the-art accuracy in estimating affective lexicons.

## 2 BACKGROUND

In this section, we start with a review of ACT and how it uses sentiment of social characters to model interactions. Then we review related works to estimate affective meanings from corpus and discuss their main limitations. Finally, we briefly review the BERT model and how we can use this model as a deep embedding space.

### 2.1 AFFECT CONTROL THEORY

According to ACT, every concept representing interactions is quantified in EPA space. The baseline EPA representation of words is known as *sentiments*. Figure 1 visualizes EPA representation of some sample words. In this plot, we can observe that "suicidal" and "nervous" both have bad, powerless, and passive meanings but "suicidal" is more negative in all three dimensions. On the other hand, "happy" is a pleasant, powerful, and active word. Note the range of EPA ratings is from -4.3 to 4.3.

ACT considers social interactions or events that include an *actor* that *behaves* toward an *object*. Extracting the Actor-Behavior-Object (ABO) components of an event is the first step in modeling interactions (Heise, 2010). Actor/Object in an event has an identity such as "baby" or "boss". In some cases, the characteristics of an actor/object is part of the identity. For example, the identity of a person is "nervous boss". In these cases, modifiers are amalgamated to their identities. Figure. 1 is a visualization of sample words in EPA space. It uses symbols to represent that a concept is evaluated as identity, behavior, or modifier.

According to ACT theory, social events make *impressions* on ABO characters (Robinson & Smith-Lovin, 1999). Consider an example of observing "a bossy employer argues with an employee". This observation leads people to evaluate both the *actor* and the *object* of the interaction are less pleasant than initial thoughts. They may also feel the employer is more powerful, and the employee is more powerless than their baseline sentiments after observing this event. Being more pleasant/powerful is *impression* of observing and event and it means higher value in *evaluation*/*potency*.

If actor/object behaves as expected, then *impression* of identity does not change far from baseline, but if actor/object does something unexpected, then a large change from the baseline is expected. *Deflection* is the euclidean distance between the baseline sentiments of ABO characters and their impressions following an event. If the impression of an ABO event is close to initial sentiments, deflection is small, and it gets bigger when the impression of the event leaves the initial sentiments. ACT discusses that minimizing *deflection* is the driving force in human activity. Highly deflecting events create social and physiological distress (Goldstein, 1989). For example, if a grandmother fights with her grandchild, the grandmother and the grandchild feel distressed and prefer to do something. We may expect one side to take an action, like apologize, to bring the impression of their identities back to where they views themselves in the society. This highly deflected event is very different from two soldiers fighting in a battle. The soldiers are supposed to fight with enemies in battle, so they may not feel social pressure to change their behavior.

Heise (2013) developed a software called INTERACT. It simulates sequential interactions between two identities and finds the behavior that minimizes the deflection. It can also predict attributes and emotions during the interaction. Consider the following set of events/interactions that we simulate using INTERACT,

1. Employee greets bossy employer.
2. Bossy employer asks employee.
3. Employee replies to bossy employer.
4. Bossy employer argues with employee.
5. Employee listens to / disobeys bossy employer.

The visualization in Figure 2 shows how the impression of actor/object's identity changed based on sequential interactions. Let's focus on evaluation dimension for the employer. Employer has a negative baseline evaluation but after observing the first two interactions, it increases. The first two interactions include positively rated behaviors. After second interaction, impression of the employer's identity is positively evaluated and so the next positively evaluated action, replies to, does not move it substantially. A positive behavior is expected from a positive identity. However,in the fourth interaction, the employer is evaluated to have an unpleasant identity after doing a negative behavior, argue with. For the fifth event, we have shown how the impression of different actions by the employee has significantly moved the states for both the actor and the object. The sequential interactions discussed here are similar to our mentorship example discussed the beginning. It shows how understanding the interaction dynamic can make us predict the consequences of our behavior.

ACT has rules to describe how impression of an events changes affective meaning of ABO characters. ACT uses either mathematical equations or descriptive forms to discuss these rules. The following two descriptive forms show how the identity of the *actor* is impressed by some events,

- *Actors* seem nice when they behave in a positive way toward others. This describes *morality* fact in ACT literature. Observing the *evaluation* dimension for the *actors* after he greets the *object*, we can find this *behavior* resulted in an impression of being nicer (getting larger evaluation) comparing to the state in the last step.

- Active *behaviors* make the *actors* seem more active. Observing the boss's activity, he is considered more active after he argues with [active behavior] the employee.

As we have seen in the descriptive forms, events can change the impression of ABO characters. They move them toward or away from their currentor initial sentiments.

To formulate the process in mathematical space, we briefly review the quantification process using surveys. The first step is quantifying the *sentiments* that are introduced as identity, behavior,

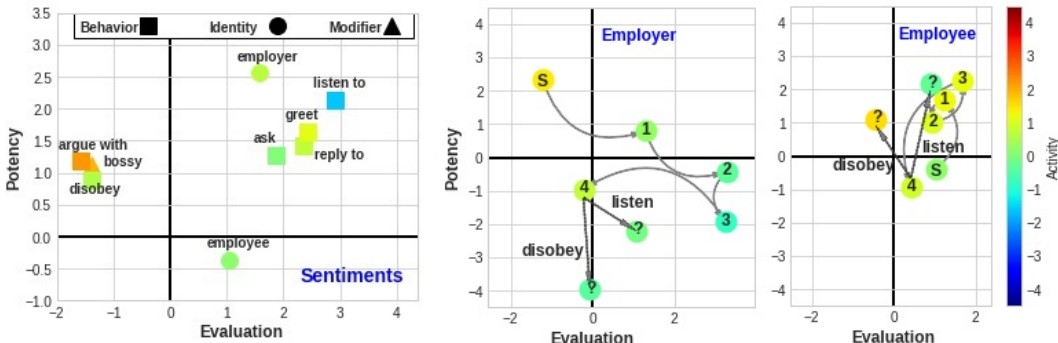

Figure 2: Simulating sequential events in an interaction between an employee and employer. The initial sentiments for both characters are shown by star signs and the sentiments after each of four events are shown by numbers. After the fourth event, based on what the employee does, the final sentiment for the two characters would be one of the places shown by the question mark.

and modifier. For this purpose, at least 25 participants rate words of interest in EPA space (Heise, 2010). In this survey, the participants rate how they feel about an identity/behavior/modifier such as "employer". After aggregating the *sentiment* surveys, every concept is assigned to its baseline affective meaning. The next step is identifying contributing facts that derive the impressions of events (Robinson & Smith-Lovin, 1999). For this purpose, the participants rate ABO characters again after observing a set of events. For example, participants rate affective meaning of "employee", "greet", and "employer" after observing "the employee greets the employer". As we discussed earlier, the ratings of ABO could be different from the initial basements. ACT uses regression models, known as impression change equations, to estimate these changes (Heise, 2013). Let $X = [A_e, A_p, A_a, B_e, B_p, B_a, O_e, O_p, O_a]^T$ represent the EPA values/sentiments of an ABO triple, where $\{A, B, O\}$ represent the ABO characters and $\{e, p, a\}$ the EPA components. Consider further the two-way interactions $X^2 = [A_eB_e, A_eO_e, A_eB_a, \ldots A_aO_a]^T$ and three-way interactions $X^3 = [A_eB_eO_e, A_eB_eO_p, A_eB_eO_a, \ldots A_aB_aO_a]^T$. The basic structure of an impression change equation is the linear model

$$X' = \alpha X + \beta X^2 + \gamma X^3 \tag{1}$$

where $\alpha, \beta$, and $\gamma$ are coefficient vectors and $X'$ represent the resulting impression after the event. Modifiers can incorporated prior to impression change by changing the baseline values/sentiments (e.g. bossy employer). Averett & Heise (1987) defined *amalgamation equations* similar to (2) to find the sentiments for an identity with a modifier.

$$A = \rho + \theta M + \psi I, \tag{2}$$

$$\rho = \begin{bmatrix} -0.17 \\ -0.18 \\ 0 \end{bmatrix}, \quad \theta = \begin{bmatrix} 0.62 & -0.14 & -0.18 \\ -0.11 & 0.63 & 0 \\ 0 & 0 & 0.61 \end{bmatrix}, \quad \psi = \begin{bmatrix} 0.50 & 0 & 0 \\ 0 & 0.56 & 0.07 \\ 0 & -0.05 & 0.60 \end{bmatrix}.$$

where, $A$, $M$, $I$, represent affective meaning of actor's identity, modifier, baseline identity and $\rho$, $\theta$, $\psi$ are vector of intercepts and coefficient matrices. Equation (2) is a weighted average of the evaluation for the modifier and the identity.

## 2.2 Finding affective meaning from text

Alhothali & Hoey (2017) used graph-based sentiment lexicon induction methods to find affective lexicons associated with words. They used similarity graphs to expand affective meanings to neighbor words in four different embedding spaces. They found that using both semantics and distributional-based approaches gives the best semi-supervised result. Li et al. (2017) argued that word-embedding can represent the words' general meaning, including denotative meaning, connotative meaning, social meaning, affective meaning, reflected meaning, collocative meaning, and thematic meaning. However, ACT uses only affective meanings of the words. So word-embedding graph propagation that uses general meaning similarities reduces accuracy for finding affective meaning.

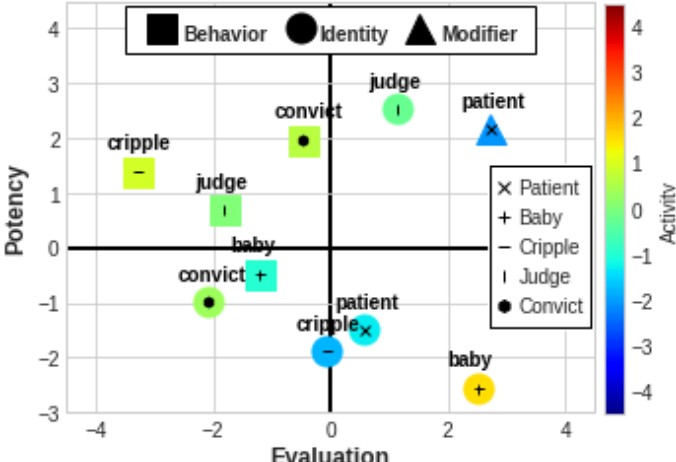

Figure 3: Visualization of words with different affective meaning in EPA space. Circles, squares, and triangles show identities, behaviors, and modifiers. The color represents the activity dimension.

Mostafavi & Porter (2021); Li et al. (2017) used supervised methods on shallow word-embeddings to find affective meanings of the words. However, all these approaches are limited by using one representation for different meanings of a word and not considering the context.

Shallow word embeddings have only one representation for every word. On the other hand, the affective meaning of a word in *identity, modifier*, or *behavior* categories are different. For example, "mother", "coach", and "fool" have very different affective meanings when they are *behavior* or *identity*, but these words have only one representation in shallow embedding space. Figure 3 shows EPA values for some words that appear in different categories. We can observe how some words are mapped very differently based on their category. For example, "baby" as an *identity* is a pleasant and active character but is unpleasant and passive as a *behavior*.

To get a sense of how similar the affective meanings are between categories, we calculated the pairwise correlation of EPA values for words shared across various different categories (Table 1). We observe that, while some categories maintain high association across categories, other categories, like *identity* and *behavior* in the *activity* and *potency* dimensions have low association implying they words used in different categories represent different affective meanings. From these tables, we observe, *activity* and *potency* dimensions of common words between *identity* and *behavior* are too small to assume they represent the same affective meanings. This suggest that ACT can benefit from models that can represent contextual aspects and differentiate between different meanings of a word.

Table 1: Correlation between the affective meaning of words in identity, behavior, and modifier category of a dictionary collected in 2014 (Smith-Lovin et al., 2016).

| | Identity-Modifier | | | | Modifier-Behavior | | | | Identity-Behavior | | |
|---|---|---|---|---|---|---|---|---|---|---|---|
| | E | P | A | | E | P | A | | E | P | A |
| E | 0.93 | 0.49 | -0.58 | E | 0.98 | 0.92 | -0.32 | E | 0.73 | 0.35 | 0.40 |
| P | 0.77 | 0.62 | -0.39 | P | 0.85 | 0.80 | -0.25 | P | 0.29 | 0.55 | 0.02 |
| A | -0.45 | 0.33 | 0.98 | A | -0.27 | -0.30 | 0.67 | A | -0.11 | 0.30 | 0.40 |

We can observe in Table 1 that affective meanings of the words in different categories are not necessarily highly correlated. For example, there is only a 0.4 correlation between *activity* dimensions of words that appear both in *identity* and *behavior* categories.

## 2.3 BIDIRECTIONAL ENCODER REPRESENTATIONS FROM TRANSFORMERS

In 2018, Google open-sourced BERT as the state-of-the-art model for a wide range of Natural Language Processing (NLP) tasks (Devlin et al., 2018). Unlike models that use previous words to predict the target word, BERT uses words on both sides of the target word in all layers. BERT gives state of

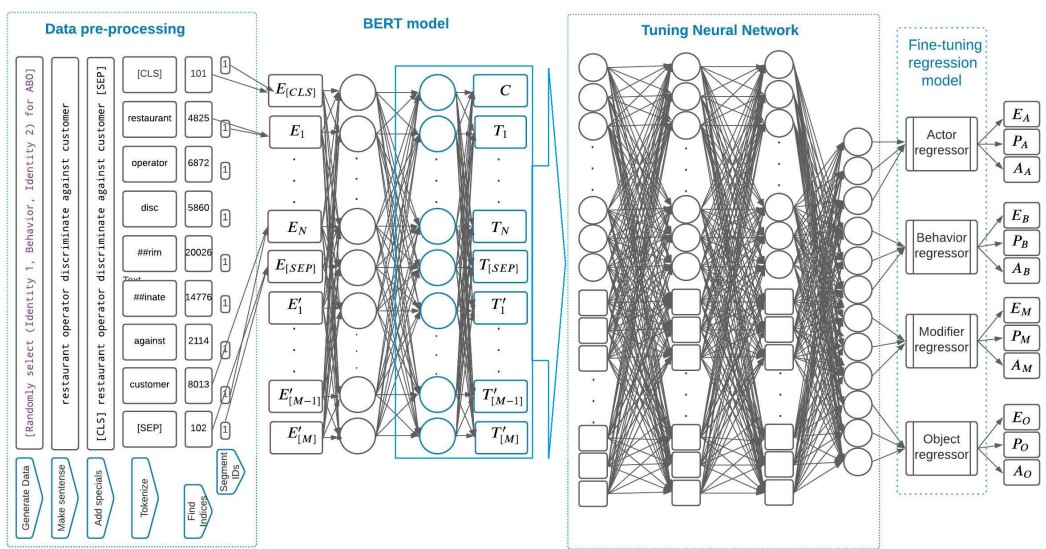

Figure 4: BERTNN pipeline. Sentences describing an event are generated in the first step and pre-processed to pass to a BERT model. Then the last two layers of the BERT model are concatenated to make contextual word-embedding. This embedding is passed to a three-layer neural network to find affective meaning, and finally, the output of the neural network is fine-tuned using a regression model.

the art in many challenging NLP tasks (McCormick & Ryan, 2019). BERT has been pre-trained on Wikipedia and book corpora that includes more than 10000 books. This pre-trained model knows how to represent texts. We use the base pre-trained version of BERT in this study to find the embedding of a sentence that describes one social event. To use BERT as a contextual embedding we need to perform the following processing, shown in Figure 4 (McCormick & Ryan, 2019).

1. Input variables for BERT model are two sequences of numbers known as Token and Segment IDs. By default, BERT assumes these sequences represent two sentences and two special tokens to indicate their relationship. The [CLS] token indicates the start of the first sentence, and the special token [SEP] comes at its end. We use BERT with only one sentence, but we have to use these special tokens.

2. The BERT tokenizer, known as WordPiece, tokenizes the input sentences. If the sentence tokens are in the vocabulary of the pre-trained model, they appear in the tokenized list without any modification. However, WordPiece assigns multiple tokens to a word if the word is not in its vocabulary. In that case, tokens are root vocabulary and suffix of the original word. For example, if the words "affective" and "subtext" are not included in the vocabulary, WordPiece outputs [affect, ##ive] and [sub, ##text] tokens where ## shows the two tokens came from a compound word.

3. Token IDs are indices of the tokens in the BERT tokenizer vocabulary. Segment IDs associate the tokens to one of the two sentences. Since we are passing only one sentence, the segment ID is set to 1 for all the tokens in the sentence.

The main advantage of word-embeddings derived from BERT over shallow word-embeddings is that BERT can take into account the context of a word. This means that words can be automatically given different representations when the words is used a as a *behavior* or an *identity*. To take advantage of BERT word embedding, we should train it on synthetic data that represents the concepts and their category simultaneously.

Table 2: Comparing different architectures for the neural network. HL denotes the number of hidden layers in the network.

| HL | Iterations | Neurons | Train Loss | Val Loss |
|---|---|---|---|---|
| 3 | 400 | NL + Linear | **0.24** | **0.22** |
| 3 | 400 | Linear | 1.87 | 1.68 |
| 3 | 2000 | Linear | 1.63 | 1.73 |
| 2 | 400 | NL + Linear | 0.33 | 0.31 |
| 2 | 2000 | NL + Linear | 0.29 | 0.29 |
| 1 | 400 | NL + Linear | 0.49 | 0.47 |
| 1 | 2000 | NL + Linear | 0.42 | 0.42 |

## 3 METHOD

We introduce a new framework, BERTNN, that processes synthetic data, passes it to a pre-trained BERT model, and fine-tunes the result to generate extended affective meaning dictionaries. The pipeline for our method (BERTNN) is shown in Figure 4 and we discuss the details in this section.

The data used to train our model was generated from the affective dictionary described in Smith-Lovin et al. (2016). This dictionary was developed from surveys conducted between 2012-2014 and includes 929 *identities*, 814 *behaviors*, and 660 *modifiers*. To partition the data into training and test set, we used stratified sampling from affective dictionaries. The three categories of *identities*, *behaviors*, and *modifiers* in the affective dictionaries are one "strata". We randomly sample 85% of the words in each strata for the training and the remaining 15% are selected as the test set.

To generate each sample of the synthetic data, two identities, one modifier, and one behavior are randomly selected from the training set. Sentences with an *Actor Behaves Modified Actor* (ABMO) grammar is made (e.g., Employee greets bossy employer). In other words, each sample describes an event in ABMO grammar. We used 10000 sample events to train the regression model. After pre-processing and tokenizing ABMO sentences similar to "data pre-processing" part of the pipeline, they are sent to the pre-trained BERT model.

The mechanism of defining sentence embedding from the BERT model has a significant impact on performance. We tried mechanisms such as using the values in the last layer, finding the mean values from the last two layers, and concatenating the last two layers. The best result came from deep embedding formed by concatenating the last two layers.

The next step is finding a mapping from the sentence embedding to the affective space. Mostafavi & Porter (2021) used interactions between the features for better affective predictions. We used a four-layer neural network to provide a non-linear mapping from the BERT output to the 12-dimensional ABMO values. In this neural network, the first three layers have 12 *dense* and 12 *Tanh* parts. Here, we used hyperbolic tangent functions are used as the nonlinear functions in the network. The last layer is a dense layer that produces the 12-dimensional vector. The $L_2$ loss used in the neural networks minimized the squared error between estimated affective meaning and the target values across all 12 dimensions. However, every ABMO character has a 3-dimensional representation independent of other characters in ABMO grammar. In this case, considering four different regression models assigns a loss function for each character and fine-tunes its values. In short, linear regression models fine-tune the neural network output to find affective meanings of all the characters.

We implemented the neural network in Python using Trax package. We used stochastic gradient descent with a learning rate of 0.01 and batch size of 50. Table 2 shows the $L_2$ loss for several different architectures and iterations. The network with only linear neurons has 12 neurons in the hidden layers. When nonlinear (NL) neurons are used, the hidden layers have 12 dense neurons and 12 hyperbolic tangent neurons. We found that the minimal $L_2$ loss is achieved in 400 iterations with 3 hidden-layers and NL neurons.

The neural network output is a 12-dimension vector that is highly correlated with the affective meaning variables. However, Root Mean Square Error (RMSE), Mean Absolute Error (MAE) of the network output was improved with a regression layer. We implemented ordinary least squares Linear

Table 3: Performance of several models on (Smith-Lovin et al., 2016) data. Bold indicates the best model. Our model, BERTNN, performed best in most categories.

|  |  | MAE | | | RMSE | | | Correlation | | |
|---|---|---|---|---|---|---|---|---|---|---|
|  |  | E | P | A | E | P | A | E | P | A |
| **Identity** | Analogy stepW. | 0.93 | 0.92 | 0.81 | 1.2 | 1.13 | 0.99 | 0.65 | 0.53 | 0.18 |
|  | Analogy_regression | 0.95 | 0.95 | 0.81 | 1.22 | 1.16 | 1.00 | 0.64 | 0.48 | 0.15 |
|  | StepW Translation | 0.58 | 0.63 | 0.57 | 0.80 | 0.83 | 0.72 | 0.85 | 0.78 | 0.68 |
|  | BERTNN | **0.42** | **0.38** | **0.28** | **0.51** | **0.47** | **0.37** | **0.94** | **0.93** | **0.93** |
| **Behavior** | Analogy stepW. | 1.17 | 0.71 | 0.68 | 1.44 | 0.90 | 0.83 | 0.73 | 0.45 | 0.51 |
|  | Analogy_regression | 1.20 | 0.75 | 0.73 | 1.48 | 0.93 | 0.90 | 0.71 | 0.44 | 0.38 |
|  | StepW Translation | 0.80 | 0.56 | 0.58 | 1.02 | 0.74 | 0.73 | 0.87 | 0.66 | 0.67 |
|  | BERTNN | **0.59** | **0.41** | **0.27** | **0.75** | **0.5** | **0.36** | **0.94** | **0.86** | **0.93** |
| **Modifier** | Analogy stepW. | 0.33 | 0.73 | 0.88 | 1.08 | 0.9 | 1.09 | 0.87 | 0.86 | 0.57 |
|  | Analogy_regression | 0.56 | 0.76 | 0.92 | 1.16 | 0.93 | 1.10 | 0.86 | 0.86 | 0.57 |
|  | StepW Translation | **0.32** | 0.53 | 0.57 | 0.86 | 0.67 | 0.70 | 0.92 | 0.91 | 0.85 |
|  | BERTNN | 0.50 | **0.39** | **0.38** | **0.60** | **0.49** | **0.44** | **0.96** | **0.95** | **0.94** |

Regression using the Sklearn package in Python. We can share the code upon request. Also, the data used in this project is publicly available on the internet.

The tuning layers in our framework are trained based on the target affective dictionary. After training, this pre-trained model can predict affective meaning for new concepts. The user should make an ABMO event that includes the concept of interest and pass it to the model, and the affective meaning would be the output of the network. For example, if "moderator" is a new *identity* outside the affective dictionary, we make an event that includes this identity as the *actor* or *object* such as, "moderator help angry client". The event is passed to a pre-trained model of this project, then the output of actor regressor is the predictive affective meaning for "moderator" identity. Here the affective meaning that we get for one specific *identity* as *actor* or *object* is the same, so it does not change anything if we consider the *identity* of interest as *actor* or *object*.

## 4  RESULTS

We compared the performance of our model with different word analogy (Kozlowski et al., 2019), regressions (Li et al., 2017), and translation matrix methods (Mostafavi & Porter, 2021) to find affective meanings (Table 3). We used RMSE, MAE, and correlation analysis to compare the result. The test data used to compare these methods included 139 *identities*, 122 *behaviors*, and 99 *modifiers* came from stratified sampling discussed earlier.

All the similar works used shallow embedding (Kozlowski et al., 2019; Mostafavi & Porter, 2021; Li et al., 2017). We used the publicly available code of Mostafavi & Porter (2021) and Kozlowski et al. (2019) to replicate their work and compare the result. For Li et al. (2017) we had to implement their method in Python. To further improve their methods, we added tuning layers such as adding step-wise regression to the analogy method. Table 3, shows the best result we could get from other methods. We can observe from this table that our approach reached state of the art across most of the metrics.

To evaluate how close our extended dictionaries are to the baseline affective meanings, we calculated the correlation for *identity* and *behaviors* in Tables 5 and 6. The correlation (a) between the result of the test set in our method, (EE, EP, and EA) and the EPA values from the surveys (E, P, and A) are shown. The diagonal terms are the correlation values we have seen earlier in Table 3. Also, you can find the correlation between the three dimensions from the survey data are shown in (b), and the correlation between estimated three dimensions are shown in (c). and survey results. We can observe in both tables the values in tables (b) and (c) are close. It reveals that BERTNN estimation is highly correlated with survey data and the cross-term dynamics is well estimated.

One problem with estimation from shallow-embeddings was having the same embedding for words as *identity* or *behavior*. Using the last two layers of the BERT model as embedding, we can differ-

Table 4: Comparing estimated affective meanings for "judge" as an *identity* and *behavior*.

| *Judge* | Evaluation | Potency | Activity | Est. Evaluation | Est. Potency | Est. Activity |
|---|---|---|---|---|---|---|
| **Behavior** | -1.83 | 0.71 | 0.07 | -2.03 | 0.98 | 0.24 |
| **Identity** | 1.15 | 2.53 | -0.22 | 1.57 | 1.56 | -0.42 |

Table 5: Correlation analysis for identities. (a) Correlation between estimated values (EA, EP, and EA) and the affective dictionary value (E, P, and A). We can also compare the correlation of the words in the three dimensions shown in (b) and correlation of the estimated values of the three dimension shown in (c) to find how close the off-diagonal entries are in the estimation comparing to dictionary values.

| (a) | E | P | A |
|---|---|---|---|
| **EE** | 0.93 | 0.57 | 0.06 |
| **EP** | 0.57 | 0.93 | 0.31 |
| **EA** | 0.04 | 0.25 | 0.93 |

| (b) | E | P | A |
|---|---|---|---|
| **E** | 1.00 | 0.55 | 0.05 |
| **P** | 0.55 | 1.00 | 0.25 |
| **A** | 0.05 | 0.25 | 1.00 |

| (c) | EE | EP | EA |
|---|---|---|---|
| **EE** | 1.00 | 0.62 | 0.06 |
| **EP** | 0.62 | 1.00 | 0.33 |
| **EA** | 0.06 | 0.33 | 1.00 |

entiate between these two cases. In Table 4 we can observe that estimated values for "judge" are different when it is considered as *identity* or *behavior*.

Tables 5 and 6 show that diagonal terms in the correlation of estimated values and values from the survey dictionary are reasonably large. On the other hand, the off-diagonal entries from the estimation are close to the ones from the survey.

## 5 CONCLUDING REMARKS

A big limitation in the applicability of ACT to different contexts such as NLP tasks was the limitation of affective vocabulary collected from surveys. In this paper, an approach to make training and test sentences that describe an ABMO event. Then we used BERT as an embedding and fine-tuned it using a neural network and regression model. Our approach, BERTNN, resulted in state of the art in estimating affective meaning.

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

Table 6: Correlation analysis for behaviors. (a) Correlation between estimated values (EA, EP, and EA) and the affective dictionary value (E, P, and A). We can also compare the correlation of the words in the three dimensions shown in (b) and correlation of the estimated values of the three dimension shown in (c) to find how close the off-diagonal entries are in the estimation comparing to dictionary values.

| (a) | E | P | A |
|---|---|---|---|
| **EE** | 0.94 | 0.55 | -0.23 |
| **EP** | 0.59 | 0.86 | 0.13 |
| **EA** | -0.25 | 0.11 | 0.93 |

| (b) | E | P | A |
|---|---|---|---|
| **E** | 1.00 | 0.59 | -0.21 |
| **P** | 0.59 | 1.00 | 0.12 |
| **A** | -0.21 | 0.12 | 1.00 |

| (c) | EE | EP | EA |
|---|---|---|---|
| **EE** | 1.00 | 0.63 | -0.26 |
| **EP** | 0.63 | 1.00 | 0.12 |
| **EA** | -0.26 | 0.12 | 1.00 |

David R Heise. Social action as the control of affect. *Behavioral Science*, 22(3):163–177, 1977.

David R Heise. *Surveying cultures: Discovering shared conceptions and sentiments*. John Wiley & Sons, 2010.

David R Heise. Interact guide. *Department of Sociology, Indiana University*, 2013.

Austin C Kozlowski, Matt Taddy, and James A Evans. The geometry of culture: Analyzing the meanings of class through word embeddings. *American Sociological Review*, 84(5):905–949, 2019.

Darys J Kriegel, Muhammad Abdul-Mageed, Jesse K Clark, Robert E Freeland, David R Heise, Dawn T Robinson, Kimberly B Rogers, and Lynn Smith-Lovin. A multilevel investigation of arabic-language impression change. *International Journal of Sociology*, 47(4):278–295, 2017.

Minglei Li, Qin Lu, Yunfei Long, and Lin Gui. Inferring affective meanings of words from word embedding. *IEEE Transactions on Affective Computing*, 8(4):443–456, 2017.

Chris McCormick and Nick Ryan. Bert word embeddings tutorial, May 2019. URL `http://www.mccormickml.com`.

Moeen Mostafavi. Adapting online messaging based on emotiona. In *Proceedings of the 29th Conference on User Modeling, Adaptation and Personalization*, 2021.

Moeen Mostafavi and Michael Porter. How emoji and word embedding helps to unveil emotional transitions during online messaging. In *2021 IEEE International Systems Conference (SysCon)*. IEEE, 2021.

Charles Egerton Osgood, George J Suci, and Percy H Tannenbaum. *The measurement of meaning*. Number 47. University of Illinois press, 1957.

Julie M Robillard and Jesse Hoey. Emotion and motivation in cognitive assistive technologies for dementia. *Computer*, 51(3):24–34, 2018.

Dawn T Robinson and Lynn Smith-Lovin. Selective interaction as a strategy for identity maintenance: An affect control model. *Social Psychology Quarterly*, pp. 12–28, 1992.

Dawn T Robinson and Lynn Smith-Lovin. Emotion display as a strategy for identity negotiation. *Motivation and Emotion*, 23(2):73–104, 1999.

Dawn T Robinson and Lynn Smith-Lovin. Affect control theories of social interaction and self. 2018.

Kimberly B Rogers and Lynn Smith-Lovin. Action, interaction, and groups. *The Wiley Blackwell Companion to Sociology*, pp. 67–86, 2019.

Lynn Smith-Lovin, Dawn T Robinson, Bryan C Cannon, Jesse K Clark, Robert Freeland, Jonathan H Morgan, and Kimberly B Rogers. Mean affective ratings of 929 identities, 814 behaviors, and 660 modifiers in 2012-2014. *University of Georgia: Distributed at UGA Affect Control Theory Website: http://research. franklin. uga. edu/act*, 2016.

