# OpenReview forum: "Learning affective meanings that derives the social behavior using Bidirectional Encoder Representations from Transformers"
_ICLR.cc/2022/Conference — ICLR 2022 Submitted_

### Official Review · Reviewer_Ketb · 2021-10-30

**Correctness:** 3
**Technical Novelty And Significance:** 2
**Empirical Novelty And Significance:** 3
**Recommendation:** 5
**Confidence:** 3

**Main Review:**

This paper proposes a new sentiment representation method by using Affect Control Theory (ACT) and BERT model. Using ACT could capture the social interactions and sentiment changes. The main contribution of this paper is the usage of ACT and the whole process. According to the experimental results, the BERT model shows good performance.

+Ves
+ The usage of Affect Control Theory is very interesting and the first work.
+ The paper is a nice effort with visualizations and nice structure.

-Concerns

-The ablation studies and some experiments should be expanded. For instance, changing different numbers of NN layers, adding other word embedding comparative experiments to show the effectiveness of BERT model.

-Many experimental details are insufficient, such as parameters.

-How to use this model to extend affective dictionaries? The authors should add some examples or explain clearly.

-For the input dataset, how to distinguish the different categories of words, like identity or behavior.

-In Method, the authors use One affective dictionary, which dictionary, please give the reference.

-When a word is polysemous, like apple (fruit or company), how to do in this model.

Minor comments:

*The figures are a bit blurry, please replace them with vector diagrams.


**Summary Of The Paper:**

This paper proposes a new sentiment representation method by using Affect Control Theory (ACT) and BERT model. Using ACT could capture the social interactions and sentiment changes. The main contribution of this paper is the usage of ACT and the whole process. According to the experimental results, the BERT model shows good performance.

**Summary Of The Review:**

The idea is interesting and results show the effectiveness. However, something is not clear and some ablation studies are missing.

---

> ### Author Response · Authors · 2021-11-17
> **Author response to Reviewer Ketb (part b)**
>
> __In Method, the authors use One affective dictionary, which dictionary, please give the reference.__
>
> The reference is provided in the method section:
> The data used to train our model was generated from the affective dictionary described in reference [*]. This dictionary was developed from surveys conducted between 2012-2014 and includes 929 identities, 814 behaviors, and 660 modifiers. To partition the data into training and test set, we used stratified sampling from affective dictionaries.
>
> [*] Smith-Lovin, L., Robinson, D. T., Cannon, B. C., Clark, J. K., Freeland, R., Morgan, J. H., & Rogers, K. B. (2016). Mean affective ratings of 929 identities, 814 behaviors, and 660 modifiers by university of georgia and duke university undergraduates and by community members in durham, nc, in 2012-2014. University of Georgia: Distributed at UGA Affect Control Theory Website: http://research. franklin. uga. edu/act.
>
>
> __When a word is polysemous, like apple (fruit or company), how to do in this model.__
>
> There is a great discussion in the ACT community about polysemous words. For example, a doctor may mean a medical doctor or someone with a Ph.D. When the survey data is collected, one representation for all the meanings as an identity is considered. So the respondents might have different meanings in their mind.
>  In the case that we have modifiers or institutions described, we face less uncertainty. In this research, we wanted to generalize the survey results as they were collected. We did not want to cover what they had not differentiated in the first place. However, in future work that we may collect the data ourselves, we can consider contextual aspects to deal with multiple meanings.
>
> __Minor comments: *The figures are a bit blurry, please replace them with vector diagrams.__
>
> All the images and tables are revised to make them clear.

---

> ### Author Response · Authors · 2021-11-17
> **Author response to Reviewer Ketb  (part a)**
>
> We appreciate your thoughtful comments. We have revised the draft entirely, and we tried to address all your concerns in the latest draft.
>
> __The ablation studies and some experiments should be expanded. For instance, changing different numbers of NN layers, adding other word embedding comparative experiments to show the effectiveness of the BERT model.__
>
> We used several different architectures and iterations,
> “Table 2 shows the L2 loss for several different architectures and iterations. The network with only linear neurons has 12 neurons in the hidden layers. When nonlinear (NL) neurons are used, the hidden layers have 12 dense and 12  hyperbolic tangent neurons. We found that the minimal L2 loss is achieved in 400 iterations with three hidden-layers and NL neurons. ”
> All similar works used shallow word embedding and their model had one representation for a word no matter from which category it came (identity, behavior, or modifier). To evaluate our model, we implemented and compared the performance with three related approaches (Table 3). The implementation details are provided in Section 4, e.g.,
> All the similar works used shallow embedding (Kozlowski et al., 2019; Mostafavi & Porter, 2021;Li et al., 2017).  We used the publicly available code of Mostafavi & Porter (2021) and Kozlowskiet al. (2019) to replicate their work and compare the result. For Li et al. (2017) we had to implement their method in Python.  To further improve their methods, we added tuning layers such as adding stepwise regression to the analogy method. Table 3, shows the best result we could get from other methods.  We can observe from this table that our approach reached state of the art across most of the metrics.
>
> __Many experimental details are insufficient, such as parameters.__
>
> We have added more implementation detail in section 3 of the latest version:
> “To generate each sample of the synthetic data, two  identities, one modifier, and one behavior are randomly selected from the training set. Sentences with an Actor Behaves Modified Actor  (ABMO) grammar is made (e.g., Employee greets bossy employer). In other words, each sample describes an event in ABMO grammar. We used 10000 sample events to train the regression model.  After pre-processing and tokenizing ABMO sentences similar to "data pre-processing'' part of the pipeline, they are sent to the pre-trained BERT model.
> We implemented the neural network in Python using Trax package. We used stochastic gradient descent with a learning rate of 0.01 and batch size of 50.
> The neural network output is a 12-dimension vector that is highly correlated with the affective meaning variables. However, Root Mean Square Error (RMSE), Mean Absolute Error (MAE) of the network output was improved with a regression layer. We implemented ordinary least squares Linear Regression using the Sklearn package in Python. We can share the code upon request. Also, the data used in this project is publicly available on the internet. “
>
> __How to use this model to extend affective dictionaries? The authors should add some examples or explain clearly.__
>
> An explanation and example is given at the end of Section 3:
> ‘The tuning layers in our framework are trained based on the target affective dictionary. After training, this pre-trained model can predict affective meaning for new concepts. The user should make an ABMO event that includes the concept of interest and pass it to the model, and the affective meaning would be the output of the network. For example, if “moderator'' is a new identity outside the affective dictionary, we make an event that includes this identity as the actor or object such as, “moderator help  angry client''. The event is passed to a pre-trained model of this project, then the output of actor regressor is the predictive affective meaning for “moderator'' identity.’
>
> __For the input dataset, how to distinguish the different categories of words, like identity or behavior.__
>
> In generating the synthetic data, we made sentences that represent a social event. For example, “ employee shouts at bossy employer ”.  This data has ABMO grammar, and we estimate the rating for all characters in the ABMO event. We got the same affective meaning for identities as “Actor” or  “Object”, so we used one for the identity category.

---

### Official Review · Reviewer_cj4C · 2021-11-01

**Correctness:** 3
**Technical Novelty And Significance:** 2
**Empirical Novelty And Significance:** 2
**Recommendation:** 3
**Confidence:** 2

**Main Review:**

Comments on Methodology:

Firstly, the authors state they use a regression model to finetune the output of their neural network, the details of which are unclear. The authors need to elaborate on how finetuning was done with the model.

Secondly, the methodology suggested in the paper looks relatively straightforward in the following sense:  embeddings from BERT were used in a feed-forward neural network, and certain aspects of this were finetuned.

Finally, a few gaps that could strengthen the paper quite a bit are left out. For instance, the authors could also compare the model's performance using shallow word embeddings as well as contextual embeddings from advanced transformer-based architectures. They could further include the data preprocessing and implementation details for the experiments in a separate section.

Comments on Writing:

The grammar and sentence structure of many sentences in the paper could be significantly improved. Below, a few grammatical errors and sentences from the paper are listed for reference below.

 - Page 3, Line 5: "following two ones" should be "following two"

 - Page 5, Line 8: "But we have to feet the dataset so that it can understand in which category we are working." The context of this sentence is unclear.

 - Page 5, Line 10: "To use BERT as a contextual word-embedding, we should have the following processing."
The sentence may need rewording (aren't the embeddings derived from the BERT model?).

 - Page 7, Line 2: "In this paper, an approach to make training and test sentences that describe an ABO event." This line is incomplete.

 - please be consistent in wording (e.g., data-set vs dataset)

**Summary Of The Paper:**

In this paper, the authors propose a deep learning architecture to estimate the affective meaning of words in terms of their EPA (Evaluation, Potency, and Activity) scores. Their methodology achieves notable performance improvements over the approaches in the existing literature, and this is due to the contextual embeddings derived from the BERT model.

**Summary Of The Review:**

There are clear gaps in both the proposed ideas and the writing/presentation of results.

---

> ### Author Response · Authors · 2021-11-17
> **Author response to Reviewer cj4C**
>
> We appreciate your thoughtful comments. We have revised the draft entirely, and we tried to address all your concerns in the latest draft.
>
> __Firstly, the authors state they use a regression model to finetune the output of their neural network, the details of which are unclear. The authors need to elaborate on how finetuning was done with the model.__
>
> We have added more details and intuitions about using the regression model in the Method section of the revised draft.
> The L2 loss used in the neural networks minimized the squared error between estimated affective meaning and the target values across all 12 dimensions. However, every ABMO character has a 3-dimensional representation independent of other characters in ABMO grammar. In this case, considering four different regression models, assigns a loss function for each character and fine-tunes its values.  In short, linear regression models fine-tune the neural network output to find affective meanings of all the characters.
> The output of the neural network is a 12-dimension vector that is highly correlated with the affective meaning variables. However, Root Mean Square Error (RMSE), Mean Absolute Error (MAE) of the network output was improved with a regression layer. We implemented ordinary least squares Linear Regression using the Sklearn package in Python.
>
> __Secondly, the methodology suggested in the paper looks relatively straightforward in the following sense: embeddings from BERT were used in a feed-forward neural network, and certain aspects of this were fine tuned.__
>
> Considering the contextual aspects of the problem in generating synthetic data made BERT a great candidate to find contextual embedding. After considering different scenarios in selecting layers for embedding and fine-tuning, the model achieved state-of-the-art accuracy. So we expected that making the model more complicated may not contribute to the final result.
> In the dataset that we used for this project, the minimum sample size in the survey data for one specific word was 21. In other words, each concept has been rated by at least 21 participants.  David Heise  [Ref1] estimated the standard error of mean sentiment rating for a sample size of  20 is 0.36, and 0.39, 0.4 in evaluation, potency, and activity dimensions. Comparing the result we got in Table 3 with these numbers, we can find the error we got is within an acceptable range of errors.
> [Ref1]Heise, David R. Surveying cultures: Discovering shared conceptions and sentiments. John Wiley & Sons, 2010. [Table 8.5 on page 105]
>
> __Finally, a few gaps that could strengthen the paper quite a bit are left out. For instance, the authors could also compare the model's performance using shallow word embeddings as well as contextual embeddings from advanced transformer-based architectures. They could further include the data preprocessing and implementation details for the experiments in a separate section.__
>
> Thank you for these suggestions. We have updated the paper to include details of the other models we compared against (see the beginning of Section 4) and the performance of all methods in Table 3. An expanded description of the experimental data generation is provided at the start of Section 3.
>
> __Comments on Writing:__
>
> __The grammar and sentence structure of many sentences in the paper could be significantly improved.__
>
> We have carefully revised the entire paper and addressed the issues you raised.

---

### Official Review · Reviewer_UkRH · 2021-11-01

**Correctness:** 2
**Technical Novelty And Significance:** 2
**Empirical Novelty And Significance:** 2
**Recommendation:** 3
**Confidence:** 4

**Main Review:**

Strengths:

The idea of using a pre-trained model to  aid in finding a sentiment lexicon based on ACT theory is interesting.

Weaknesses:

The experimental analysis is clearly insufficient.：
-The data set used for the experiments was not presented.
-The parameters involved in the methods, such as learning rate, batch-size, etc., are not described
- No relevant work of others was cited.
-The analysis about the experimental results are too superficial.

The figures and tables of the article obviously need to be redrawn, e.g. Table 2 and Figure 4.


**Summary Of The Paper:**

 This paper uses a fine-tuned Bidirectional Encoder Representations from Transformers (BERT) model for finding affective lexicons based on ACT theory.

**Summary Of The Review:**

This paper contains too much background knowledge up front, while the experimental part is too little and the analysis is too superficial.

---

> ### Author Response · Authors · 2021-11-17
> **Author response to Reviewer UkRH**
>
> We appreciate your thoughtful review. We have revised the draft entirely, and we tried to address all your concerns in the latest draft.
>
> __The experimental analysis is clearly insufficient.：__
>
> __-The data set used for the experiments was not presented.__
>
> We have enhanced the description of the experiments. We have updated the paper to include details of the other models we compared against (see the beginning of Section 4) and the performance of all methods in Table 3. The following expanded description of the experimental data generation is provided at the start of Section 3:
> The data used to train our model was generated from the affective dictionary described in reference [*]. This dictionary was developed from surveys conducted between 2012-2014 and includes 929 identities, 814 behaviors, and 660 modifiers. To partition the data into training and test set, we used stratified sampling from affective dictionaries.
>
> [*] Smith-Lovin, L., Robinson, D. T., Cannon, B. C., Clark, J. K., Freeland, R., Morgan, J. H., & Rogers, K. B. (2016). Mean affective ratings of 929 identities, 814 behaviors, and 660 modifiers by university of georgia and duke university undergraduates and by community members in durham, nc, in 2012-2014. University of Georgia: Distributed at UGA Affect Control Theory Website: http://research.franklin.uga.edu/act.
>
> __-The parameters involved in the methods, such as learning rate, batch-size, etc., are not described__
>
> We have added more implementation detail in section 3 of the latest version:
> “To generate each sample of the synthetic data, two identities, one modifier, and one behavior are randomly selected from the training set. Sentences with an Actor Behaves Modified Actor  (ABMO) grammar is made (e.g., Employee greets bossy employer). In other words, each sample describes an event in ABMO grammar. We used 10000 sample events to train the regression model.  After pre-processing and tokenizing ABMO sentences similar to "data pre-processing'' part of the pipeline, they are sent to the pre-trained BERT model.
> We implemented the neural network in Python using Trax package. We used stochastic gradient descent with a learning rate of 0.01 and batch size of 50. Table 2 shows the L2 loss for several different architectures and iterations. The network with only linear neurons has 12 neurons in the hidden layers. When nonlinear (NL) neurons are used, the hidden layers have 12 dense and 12  hyperbolic tangent neurons. We found that the minimal L2 loss is achieved in 400 iterations with three hidden-layers and NL neurons.
> The neural network output is a 12-dimension vector that is highly correlated with the affective meaning variables. However, Root Mean Square Error (RMSE), Mean Absolute Error (MAE) of the network output was improved with a regression layer. We implemented ordinary least squares Linear Regression using the Sklearn package in Python. We can share the code upon request. Also, the data used in this project is publicly available on the internet. “
>
> __No relevant work of others was cited.__
>
> Section 2.2 now includes a description of the related literature. All these approaches used shallow word embedding and their model had one representation for a word no matter from which category it came (identity, behavior, or modifier). To evaluate our model, we implemented and compared the performance with three related approaches (Table 3). The implementation details are provided in Section 4, e.g.,
> All the similar works used shallow embedding (Kozlowski et al., 2019; Mostafavi & Porter, 2021;Li et al., 2017).  We used the publicly available code of Mostafavi & Porter (2021) and Kozlowskiet al. (2019) to replicate their work and compare the result. For Li et al. (2017) we had to implement their method in Python.  To further improve their methods, we added tuning layers such as adding stepwise regression to the analogy method. Table 3, shows the best result we could get from other methods.  We can observe from this table that our approach reached state of the art across most of the metrics.
>
> __The analysis about the experimental results are too superficial.__
>
> We have significantly expanded the results section (Section 4) to include more details and descriptions of our experimental set-up and evaluation.
>
> __The figures and tables of the article obviously need to be redrawn, e.g. Table 2 and Figure 4__
>
> All figures and tables have been updated.

---

> > ### Comment · Reviewer_UkRH · 2021-11-25
> > **Improvements**
> >
> > The latest draft does have some improvements in various aspects compared to the initial version, but I still think that the method and experiment parts of the paper need further revision. In addition, there are still some problems with the layout and structure of some figures and tables in this paper, such as Figure 4 and Table 6.

---

### Official Review · Reviewer_4E5D · 2021-11-05

**Correctness:** 2
**Technical Novelty And Significance:** 1
**Empirical Novelty And Significance:** 1
**Recommendation:** 1
**Confidence:** 4

**Main Review:**

Strengths and Weakness:
(1) The authors tackle the problem of extending affective vocabulary through a fine-tuning approach and overcoming the drawbacks of the traditional approach of vocabulary collected from surveys. Results obtained from the paper shows that the fine-tuned model achieves shows a high correlation between the estimated values and affective dictionary values for identities and behavior
(2) The method is pretty simple for fine-tuning the BERT model and is not explained properly. The authors also provide no information about the parameters used for training which makes it impossible to reproduce the results of the paper.
(3) The evaluation section is pretty weak and there are no baselines or other approaches that are being compared for this task.
(4) The data processing and preparing the training and test set to describe an ABO event is unclear and needs to be explained properly.
(5) The authors make a lot of claims that are unsubstantiated in this paper especially about extending the affective dictionaries. There are no results to demonstrate those.

**Summary Of The Paper:**

In this paper, the authors propose using BERT embedding as an alternative to shallow word embeddings to find affective lexicons and also show that fine-tuning the model achieves state-of-the-art in estimating affective meaning.

**Summary Of The Review:**

In this paper, the authors fine-tune BERT to find affective lexicons and show high correlations between estimated values and dictionary values. However, the paper is filled with a lot of drawbacks from methodology, style of writing, and results.

---

> ### Author Response · Authors · 2021-11-17
> **Author response to Reviewer 4E5D (part b)**
>
> __(3)The evaluation section is pretty weak and there are no baselines or other approaches that are being compared for this task.__
>
> We have compared our model against three related approaches (see Table 3). The implementation details are provided in Section 4, e.g.,
> All the similar works used shallow embedding (Kozlowski et al., 2019; Mostafavi & Porter, 2021;Li et al., 2017).  We used the publicly available code of Mostafavi & Porter (2021) and Kozlowskiet al. (2019) to replicate their work and compare the result. For Li et al. (2017) we had to implement their method in Python.  To further improve their methods, we added tuning layers such as adding stepwise regression to the analogy method. Table 3, shows the best result we could get from other methods.  We can observe from this table that our approach reached state of the art across most of the metrics.
>
> __(4)The data processing and preparing the training and test set to describe an ABO event is unclear and needs to be explained properly.__
>
> We have added the following description of the synthetic data generation, including the addition of including modifiers (ABMO grammar), in Section 3:
> The data used to train our model was generated from the affective dictionary described in Smith-Lovin et al. (2016). This dictionary was developed from surveys conducted between 2012-2014 and includes 929 identities, 814behaviors, and 660 modifiers.  To partition the data into training and test set, we used stratified sampling from affective dictionaries.  The three categories of identities,behaviors, and modifiers in the affective dictionaries are one “strata”. We randomly sample 85% of the words in each strata for the training and the remaining 15% are selected as the test set.
> To generate each sample of the synthetic data, two identities, one modifier, and one behavior are randomly selected from the training set. Sentences with an Actor Behaves Modified Actor (ABMO) grammar is made (e.g., Employee greets bossy employer). In other words, each sample is describing an event in ABMO grammar. We used 10000 sample events to train the regression model.  After pre-processing and tokenizing ABMO sentences similar to “data pre-processing'' part of the pipeline, they are sent to the pre-trained BERT model.
>
> __(5)The authors make a lot of claims that are unsubstantiated in this paper especially about extending the affective dictionaries. There are no results to demonstrate those.__
>
> We have substantially revised the entire draft to ensure proper clarification of how our claims are addressed.

---

> ### Author Response · Authors · 2021-11-17
> **Author response to  Reviewer 4E5D**
>
> We appreciate your thoughtful comments. We have revised the draft entirely to address all your concerns in the latest draft.
>
> __(2)The method is pretty simple for fine-tuning the BERT model and is not explained properly. The authors also provide no information about the parameters used for training which makes it impossible to reproduce the results of the paper.__
>
> We have significantly revised the entire paper to better explain all aspects of the model description and implementation. In particular, we have added more implementation detail in section 3 of the latest version:
> “To generate each sample of the synthetic data, two identities, one modifier, and one behavior are randomly selected from the training set. Sentences with an Actor Behaves Modified Actor  (ABMO) grammar is made (e.g., Employee greets bossy employer). In other words, each sample describes an event in ABMO grammar. We used 10000 sample events to train the regression model.  After pre-processing and tokenizing ABMO sentences similar to "data pre-processing'' part of the pipeline, they are sent to the pre-trained BERT model.
> We implemented the neural network in Python using Trax package. We used stochastic gradient descent with a learning rate of 0.01 and batch size of 50. Table 2 shows the L2 loss for several different architectures and iterations. The network with only linear neurons has 12 neurons in the hidden layers. When nonlinear (NL) neurons are used, the hidden layers have 12 dense and 12  hyperbolic tangent neurons. We found that the minimal L2 loss is achieved in 400 iterations with three hidden-layers and NL neurons.
> The neural network output is a 12-dimension vector that is highly correlated with the affective meaning variables. However, Root Mean Square Error (RMSE), Mean Absolute Error (MAE) of the network output was improved with a regression layer. We implemented ordinary least squares Linear Regression using the Sklearn package in Python. We can share the code upon request. Also, the data used in this project is publicly available on the internet. “
>
> We have also added more details and intuitions about using the regression model in the Method section of the revised draft.
> "The L2 loss used in the neural networks minimized the squared error between estimated affective meaning and the target values across all 12 dimensions. However, every ABMO character has a 3-dimensional representation independent of other characters in ABMO grammar. In this case, considering four different regression models, assigns a loss function for each character and fine-tunes its values.  In short, linear regression models fine-tune the neural network output to find affective meanings of all the characters.
> The output of the neural network is a 12-dimension vector that is highly correlated with the affective meaning variables. However, Root Mean Square Error (RMSE), Mean Absolute Error (MAE) of the network output was improved with a regression layer. We implemented ordinary least squares Linear Regression using the Sklearn package in Python."

---

### Decision · Program_Chairs · 2022-01-20

**Decision:**

Reject

**Comment:**

This work presents a new sentiment representation method with the use of affect control theory and BERT. Reviewers pointed out several major concerns towards the insufficient experiments and results, as well as the lack of ablation studies and related work discussion. I would like to encourage the authors to take into account the comments from reviewers to further improve their work for a stronger version for future submissions.